# The Interaction Relationship between Land Use Patterns and Socioeconomic Factors Based on Wavelet Analysis: A Case Study of the Black Soil Region of Northeast China

**Yue Wang [1], Ge Song [2],\*  and Wenying Li [1]**

1   School of Management, Shenyang Normal University, Shenyang 110034, China; ywmath@synu.edu.cn (Y.W.); duyq@synu.edu.cn (W.L.)
2   Institute of Land Management, Northeastern University, Shenyang 110169, China
\*   Correspondence: songge@wfxy.neu.edu.cn

**Abstract:** Analyzing the interaction between land use patterns (LUPs) and socioeconomic factors (SEFs) could provide a basis for regional land spatial planning and management decisions in the future. In this study, population, gross domestic product (GDP) and land use intensity were selected to explain the relationship between SEFs and LUPs. The study designed a new method of sample line acquisition for wavelet analysis, and identified the interaction grid scales of LUP changes with SEFs in 1991, 2005 and 2019 by using cross wavelet transform analysis (XWT). Wavelet transform coherent analysis (WTC) was used to reveal the interaction direction and impact strength between LUPs and SEFs. The results showed that: (1) There were two ranges of 2978–5008 m and 24,400–29,738 m in which the grid scales showing interaction between LUPs and SEFs (population, GDP and land use intensity) from 1991 to 2019 were overlapping. (2) The interaction direction between LUPs and SEFs from 1991 to 2019 was almost negative on all sample lines, while the interaction directions of the middle sample line of population and GDP from 1991 to 2019, the end sample line of GDP in 2019, and the start sample line of land use intensity in 1991 were positive. (3) Dry land, grassland and construction land were most affected by SEFs, followed by paddy fields, forest land and other land, and the least affected were water areas during 1991 to 2019. The impact of population and GDP on LUPs was reduced, while the impact of land use intensity on LUPs was increased from 1991 to 2019. Overall, population, GDP and land use intensity were the important SEFs in the changes of LUPs, and were important factors for social progress and economic development.

**Keywords:** land use patterns; socioeconomic factors; wavelet analysis; black soil region

## 1. Introduction

The formation of land use patterns (LUPs) is the spatial projection of various influencing factors that can affect the regional social and economic development and ecological environment, and is an important basis for the diagnosis of rational use of regional land [1]. With the socioeconomic development and transformation in China, the finiteness of land supply led to intensified competition in LUPs [2,3]. The urbanization rate in China increased from 26.37% in 1991 to 60.60% in 2019, which caused a series of contradictions such as the drastic change of land development and utilization patterns [4], a large loss of cultivated land resources [5], a widened gap between the development of urban and rural areas [6] and serious ecological and environmental damage [7]. The results made the LUP change increasingly drastic in China. The state has successively implemented important strategies such as "territorial spatial planning" and "ecological civilization construction", and put forward the importance of optimizing patterns of territorial spatial development, in order to meet the requirements of spatial governance of ecological civilization in the new epoch. Therefore, study of the relationship between regional LUPs and socioeconomic fac-

tors (SEFs) is of great theoretical significance and practical value for promoting sustainable regional land utilization and sustainable social and economic development.

LUPs are the result of interaction with SEFs and the natural environment, which is the projection of various factors, such as population and gross domestic product (GDP), soil type and temperature in land use space [8]. Meanwhile, LUPs have obvious effects on water-related ecosystem services [9,10], land surface temperature [11], biodiversity [12], soil erosion [13,14], and climate [15], etc. Therefore, it is necessary to investigate the change of LUPs under various SEFs and natural environment factors. Compared with natural environment factors, the impacts of SEFs on LUP changes are more rapid, suggesting that LUPs could change in a short time due to the intervention of SEFs [16–19]. These make SEFs the main reason for LUP changes. Much research has been conducted on the effect of SEFs on LUPs [20–26]. Dong et al. (2021) investigated the spatiotemporal patterns and driving factors of land use and land cover change in the China–Mongolia–Russia economic corridor, and found that SEFs were more important to the change of LUPs than natural factors [22]. Zhou et al. (2020) considered the driving factors of land use change in rural China from 1995 to 2015, and discovered that socioeconomic development was the main driving force of construction land expansion [23]. Qie et al. (2017) established a spatial-temporal human exposure model based on land use changes in Dalian City (China) at a city scale, and found that SEFs were contributors to the spatial distribution variance of land use changes [26]. Liu et al. (2020) reported the impacts of SEFs on land use changes in the Mekong River from a watershed scale over the past 40 years, and found that SEFs such as economic development and land policies were the main driving forces of land use changes [27]. Previous studies had shown that SEFs were important driving factors of LUPs. Previous results on the effects of SEFs on the changes of LUPs were not always coincident due to the difference of spatial range in various study areas [28–31]. However, SEFs generally had both positive and negative effects on LUPs, which was less studied in previous research.

In addition, the interaction relationship between LUPs and SEFs depends on spatial scale analysis [32,33]. For the choice of the spatial scale, some studies had found that the main factors affecting food security varied substantially at city, province and national scales using stepwise regression [34]. Lin et al. (2021) analyzed the production–living–ecological space conflicts of land use with different factors at the administrative boundaries, grid and integrated multi-scale [35]. The above studies could help to reveal the interaction relationship between LUPs and SEFs at different scales. However, the grid scale is mostly determined artificially, lacking a certain scientific nature. At present, wavelet analysis is the main method that solves the problem of scale-dependent characteristics. This method could explain the research subjects with small variations, and it has been used to assess the effects of climate change and SEFs on vegetation cover variations [36,37] and the relationship between rainfall and runoff [38]. These research projects could fully reveal the relationship between two specific time series. Besides, a few studies have applied wavelet analysis in spatial continuous sequence data analysis, such as selecting a straight line sample from south to north in the region [39], in order to make the selected sample more representative. Wavelet analysis could gradually achieve multi-scale refinement of localized information of temporal (spatial) frequency, and automatically could adapt to the multi-scale identification requirements of interactions between LUPs and SEFs. However, wavelet analysis has been rarely used to report the interaction relationship between LUP changes and SEFs.

The black soil region of northeast China is one of the three black soil regions in the world. Recently, the rapid urbanization and population migration in China have caused significant changes of land use in the black soil region, which could affect agricultural productive functions [40]. Thus, it is essential to investigate the reasons for LUP changes to stabilize China's main commodity grain base. The primary objectives of this study are to analyze the relationship of LUP changes with population, GDP and land use intensity using the methods of cross wavelet transform (XWT) and wavelet transform coherence

(WTC), and an interaction relationships focus on the interaction grid scales, the interaction direction and the impact strength.

## 2. Materials and Methods

### 2.1. Study Area

Bayan County, located in the hinterland of the black soil region in Heilongjiang Province, China, has been the national grain-producing county for 10 consecutive years and is an important commodity grain base in China [41]. As a suburb of Harbin, Bayan County is a world-renowned non-GM soybean base county and an ecological demonstration county in China. This study takes Bayan County (126°45′53″–127°42′16″ N, 45°54′28″–46°40′18″ E) as the study area, which has 18 towns (Figure 1). The land area in Bayan County is 3136 km$^2$. Black soil is the most widely distributed land resource in Bayan County, accounting for 53.5% of its total land area. In 2019, the population of Bayan County was 0.421 million, and its GDP was RMB 119.0 billion. In recent years, the quality of cultivated land in the region has decreased and black soil resources have been lost due to human activities [42], making the discussion on the influence of SEFs on the LUPs in Bayan County representative.

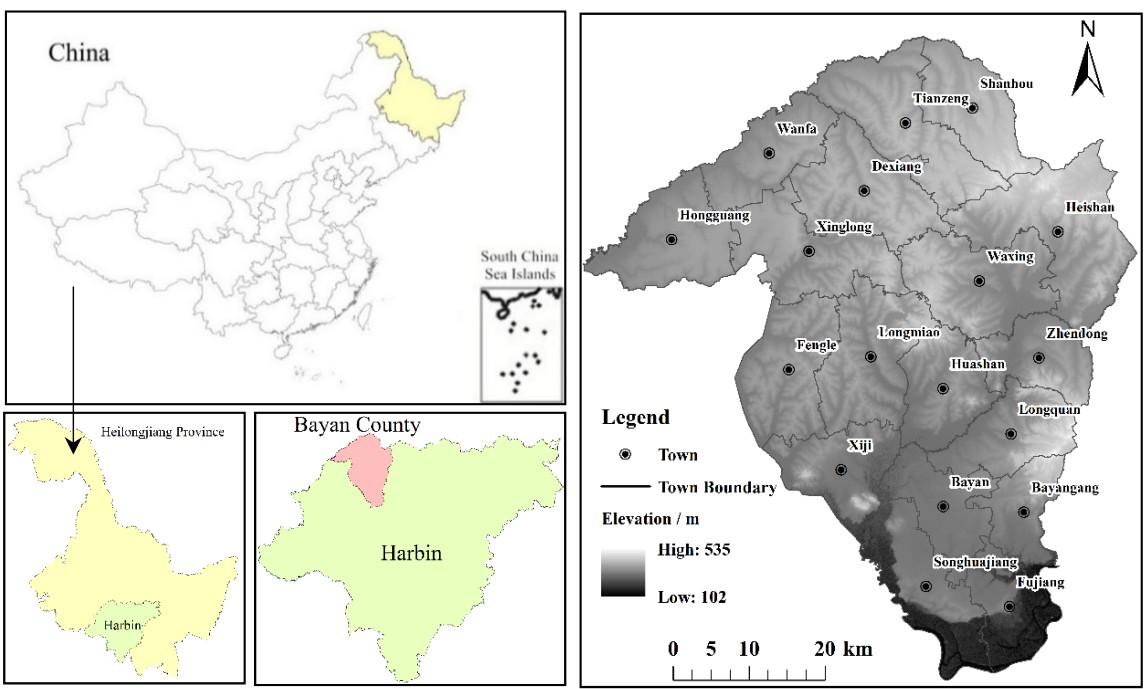

**Figure 1.** Location of the study area.

### 2.2. Research Data

#### 2.2.1. Land Use Data

This research selected the period between 1991 and 2019 as the study period of LUPs, and analyzes the relationship between LUP changes and SEFs during promulgation of the Land Administration Law Implementation Regulations in 1991 and the second revision of the China Land Administration Law in 2019. The year 2005 was used as the intermediate node for 1991 and 2019 as a reference to avoid bias caused by stochastic trends in the starting time nodes. The remote sensing images (Landsat TM/Landsat8 OLI) in the study area were downloaded from the United States Geological Survey (USGS) (http://Landsat.usgs.gov/, accessed on 21 September 2021) in 1991, 2005 and 2019, and the remote sensing image resolution was 30 × 30 m. In consideration of the natural phenological characteristics of the study area and the clarity of remote sensing images, the remote sensing images from June to September were used in the study. Cloud coverage of images was less than 10%

of the image area in the remote sensing images. The remote sensing images of the study area in 1991, 2005 and 2019 were analyzed using the software ENVI 5.1 and ArcGIS 10.2. Raster image maps for land use types with a spatial resolution of 30 m were obtained by remote sensing image processing including radiometric calibration, atmospheric correction, geometric correction and band fusion.

The LUPs were classified into seven land use types, including dry land, paddy field, forest land, grassland, construction land, water area and other land in 1991, 2005 and 2019, respectively. The area of land use types was characterized by the gradual increase of cultivated land and construction land, and by the decrease of forest land, grassland and other land from 1991 to 2019. The water area was basically relatively stable from 1991 to 2019 (Figure 2). The LUPs were covered by square grids (4148 units), and the size of a square grid was 900 × 900 m, according to Song and Wang (2017) [43], which was consistent with SEF data, and was convenient to reveal the change of LUPs on the plot scale.

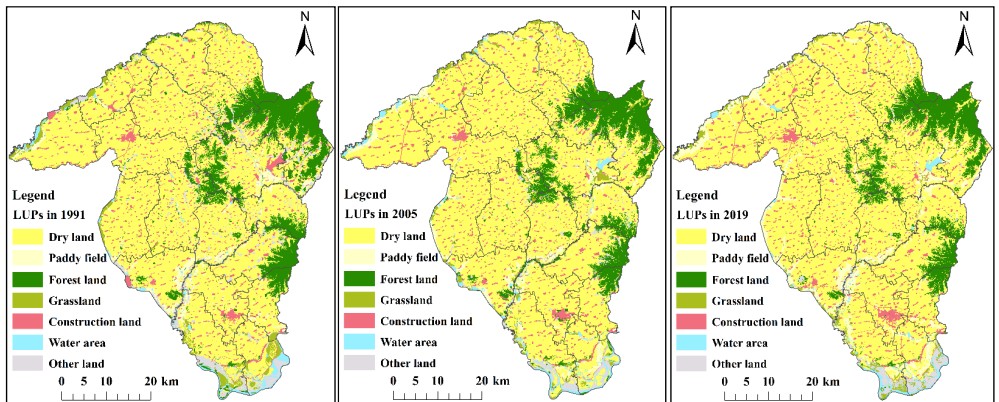

**Figure 2.** LUPs (1991, 2005 and 2019) and square grids (4148 units).

2.2.2. Data Sources and Processing of SEFs

The population and GDP spatial data in 1991, 2005 and 2019 were obtained from the Subject Database of Special Human-Land System for Informatization Construction of Chinese Academy of Sciences. In particular, the population and GDP spatial data in 2019 were derived from the "*Statistical Bulletin of National Economic and Social Development of Bayan County, Heilongjiang Province (2019)*", and the spatial data were obtained through the corrections of spatialized data in 2015. The spatial resolution of these data is 1000 m. The spatial processes of land use intensity in 1991, 2005 and 2019 were calculated by using the comprehensive index of land use intensity (formula 1), and the land use intensity was divided into other land (I), water area (II), forest and grass (III), dry land and paddy field (IV), and construction (V). In Equation (1), the land use intensity classification indexes ($i$) were set to be 1, 2, 3, 4 and 5, respectively. $A_i$ was the classification index of land use intensity class−$i$. $S_i$ was the area of land use intensity class−$i$, and $S$ was the total land area in the study area. $n$ was the classification number of the land use intensity [44].

$$L = \sum_{i=1}^{n} A_i \times (S_i/S) \tag{1}$$

The spatial distribution law of regional land use intensity is visually expressed through a mining land use intensity map based on the semi-variance function according to Fleming et al. (2015) [45]. The spatial data of population, GDP and land use intensity were converted to 900 × 900 m, in order to make them compatible with the spatial resolution of LUPs. The error created under this spatial resolution was acceptable. The calculated spatial distribution map of land use strength showed the value of land use strength was between 1.5 and 4.6, with average values of 3.68, 3.76 and 3.82 in 1991, 2005 and 2019, respectively (Figure 3).

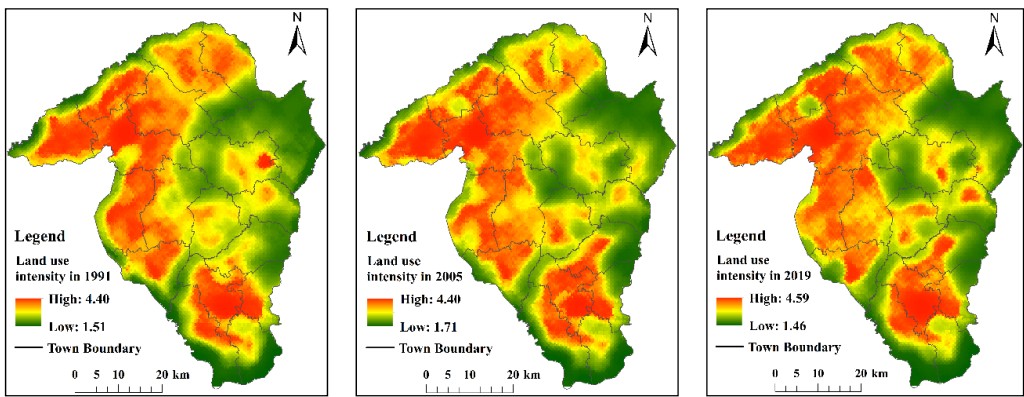

**Figure 3.** Spatial distribution characteristics of land use intensity degree.

*2.3. Model*

The basic theory of continuous wavelet transform (CWT) will be presented in the study before using cross wavelet transform (XWT) and wavelet transform coherence (WTC) analysis. XWT was used to explore the interaction grid scales between LUPs and SEFs [46]. WTC was used to measure the degree of local correlation between LUPs and SEFs. The phase angle of WTC spectrum was used to reveal the interaction direction between SEFs and LUPs.

The data input for XWT and WTC required spatially continuous data. XWT and WTC sampling line extraction should take into full account the natural conditions and the law of land use change in the study area, and include all kinds of land use types possible [47,48]. Therefore, the degree of land use diversity was used to measure the complexity of land use structure in the area.

2.3.1. Continuous Wavelet Transform (CWT)

CWT can achieve multi-scale refinement of localized information of temporal (spatial) frequency, and automatically adapt to the multi-scale identification requirements of interactions between LUPs and SEFs. The Morlet wavelet was used as the mother wavelet function (Equation (2)):

$$\Psi_0(\eta) = \pi^{-1/4} e^{i\omega_0 \eta} e^{-\eta^2/2}　　(2)$$

where $\eta$ is dimensionless spatial location and $\omega_0$ is dimensionless spatial frequency.

This study takes spatial continuous data as the data basis, and CWT decomposes spatial continuous data into low-frequency information and high-frequency information. Low-frequency information reflects the global spatial change situation of spatial continuous data, and high-frequency information reflects the change of spatial continuous data in the specific spatial position in the study area. The discrete sequence of the CWT could be defined as zooming of the signal sequence and the convolution of the normalized wavelets $x_n$ (Equation (3)).

$$W_n^X(s) = \sum_{n'=0}^{N} x_{n'} \Psi^* \left[ \frac{(n'-n)\delta t}{s} \right]　　(3)$$

where $\Psi$ is a normalized function of $\Psi_0$. $s$ is scale parameter, and $n$ is translation parameter. Equation (3) uses wavelet transformation by fast Fourier space change, which is the basis of wavelet transformation.

2.3.2. Cross Wavelet Transform (XWT)

XWT can be used to explore the interaction grid scales between LUPs and SEFs [47]. As with continuous wavelet transform (CWT), two groups of different signals (LUPs (*X*) and SEFs (*Y*)) were transformed by XWT of $W_n^X(s)$ and $W_n^Y(s)$, respectively. The smoothing spectrum operation in the spatial continuous domain was carried out by dividing the scale parameter *s*. The cross wavelet spectrum of *X* and *Y* is expressed in Equation (4), and the

corresponding cross wavelet power spectral density is shown as $|W_n{}^{XY}(s)|$. The theoretic cross wavelet power distribution of two spatial continuous signals with $P_k{}^X$ and $P_k{}^Y$ is described in Equation (5) [49].

$$W_n^{XY}(s) = W_n^X(s)W_n^{Y*}(s) \tag{4}$$

$$\frac{W_n^X(s)W_n^{Y*}(s)}{\sigma_X \sigma_Y} \Rightarrow \frac{Z_v(P)}{v}\sqrt{P_k^X P_k^Y} \tag{5}$$

In Equation (5), $\sigma_X$ and $\sigma_Y$ are the standard deviation of $X$ and $Y$, respectively. $v$ is the degree of freedom, which is determined by the signal length $n$ and the spatial delay of spatial continuous signal data. $Z_v(P)$ is the level of confidence with a probability of $P$. When the value of complex wavelet degree of freedom $v$ was 2, $Z_2$ (95%) = 3.999. When the left end of Equation (5) exceeded the confidence limit, it was considered that $X$ and $Y$ had passed the standard spectrum test of red noise under the condition of significant level $\alpha = 0.05$. The red noise power spectrum test was carried out according to Torrence and Compo (1998) [49].

### 2.3.3. Wavelet Transform Coherence (XWT)

WTC was used to measure the degree of local correlation between $X$ and $Y$. The WTC coefficient ($R_n(s)$) was used to reveal the impact strength of SEFs on LUPs. WTC spectra of two spatial continuous signal $X$ and $Y$ groups are described as Equation (6). $<>$ is a smoothing operator. $R^2{}_n(s)$ ranged from 0 to 1.

$$R_n^2(s) = \frac{\left|\langle s^{-1}W_n^{XY}(s)\rangle\right|^2}{\langle s^{-1}|W_n^X(s)|^2\rangle\langle s^{-1}|W_n^Y(s)|^2\rangle} \tag{6}$$

### 2.3.4. Phase Angle of WTC Spectrum

The phase angle of WTC spectrum was used to reveal the interaction direction between SEFs and LUPs. Under the scale of different sample lines, the leading and lagging relations between $X$ and $Y$ are described quantitatively by circular average position phase angle in Equation (7). $a_i$ was the circular position phase angle ($i = 1, \cdots, $ m), and $a_m$ was the average angle ($-\pi \le a_m \le \pi$). The phase difference between $X$ and $Y$ was not lower than 95%.

$$a_m = \arg(X, Y), \ X = \sum_{i=1}^{n}\cos(a_i), Y = \sum_{i=1}^{n}\sin(a_i) \tag{7}$$

The interaction direction between LUPs and SEFs was determined according to the direction of WTC spectrum phase angle. If the phase angle was between $\pi/2$ and $3\pi/2$ (negative), the phase change of the LUPs lagged behind the change of SEFs. That is, SEFs would lead to a change in the LUPs. If the phase angle was facing $-\pi/2$ to $\pi/2$ (forward), the phase change of LUP was ahead of SEFs. Namely, the change of LUPs would cause a change of SEFs in turn.

### 2.3.5. Sampling Method

The degree of land use diversity in the study area was measured by the Gibbs–Martin index (Equation (8)). The semi-variance function was used to excavate the map of land use diversity degree, and the spatial distribution law of regional land use diversity degree was visualized. Cambardella et al. (1994) found that when the space structure ratio in the semi-variance function was (0.75, 1], the spatial heterogeneity of land use diversity degree was caused by random factors, indicating that SEFs were the important factors. When the space structure ratio in the semi-variance function was [0, 0.25], the spatial heterogeneity of land use diversity degree was caused by structural factors, suggesting that natural factors were the key factors. When the space structure ratio in the semi-variance function was (0.25, 0.5] and (0.5, 0.75], it suggested that both SEFs and natural factors were the main impact

factors. However, natural factors and SEFs had a more distinct effect, corresponding to (0.25, 0.5] and (0.5, 0.75], respectively [50].

$$G = 1 - \left[ \sum_{i=1}^{n} x_i^2 \bigg/ \left( \sum_{i=1}^{n} x_i \right)^2 \right] \tag{8}$$

In Equation (8), G is the land use diversity index in the sample area, and its theoretical maximum $(n-1)/n$ $n$ was the number of land use types. $x_i$ was Category $i$ land area. The greater the value of $n$, the greater diversity degree of regional land use types.

On the platform of ArcGIS, the area with a high degree of land use diversity in the study area was taken as the base map. According to our previous research results (the optimal spatial analysis scale of LUPs change in the study area) [43], the spatial continuous sample blocks in the study area were selected as the sampling belt. The sampling belt was used as a mask. Extraction by mask function with ArcGIS 10.2 was used to extract the spatial raster pattern of LUPs and various SEFs in the study area, and to turn the spatial grid swatch into points. The original raster data of sample transect was converted into ASCII format. The sample line recognition algorithm was designed by using MATLAB software to identify the center line of the sample belt, and then the sampling line of LUPs in the study area was obtained. In the process of MATLAB data organization, the sample line was found to be one of the technical difficulties in this study. The study framework is presented in Figure 4.

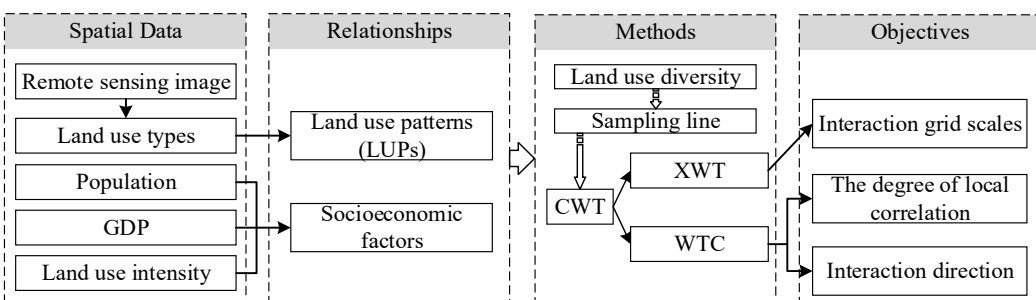

**Figure 4.** Study framework.

## 3. Results and Discussion

### 3.1. Sample Line Identification

#### 3.1.1. The Degree of Land Use Diversity

Square grids (4148 units) were used to calculate the land use diversity degree and the spatial parameters of land use diversity degree in Bayan County in 1991, 2005 and 2019, respectively, by semi-variance function (Gaussian model). The results are shown in Table 1. For 1991, the space structure ratio of the Gaussian model was 0.7662, indicating that SEFs had a significant influence on the degree of land use diversity. For 2005 and 2019, the space structure ratios of the Gaussian model were 0.7027 and 0.6824, respectively, which is within the range between 0.50 and 0.75. The results suggest that the spatial heterogeneity of land use diversity degree was caused by random factors and structural factors, and random factors were the main impact factors. SEFs and natural factors could both affect the land use diversity degree, while SEFs had more significant effects. Therefore, this study only discussed the relationship between SEFs and LUPs in the study area.

**Table 1.** Parameters of the semi-variance function model of land use diversity degree.

| Years | Model | C | $C_0$ | $C_0/(C_0 + C)$ | Error |
|-------|-------|-----|-----|-----|-----|
| 1991 | Gaussian | 0.0124 | 0.0407 | 0.7662 | 0.9567 |
| 2005 | Gaussian | 0.0165 | 0.0389 | 0.7027 | 0.9763 |
| 2019 | Gaussian | 0.0173 | 0.0377 | 0.6855 | 0.9517 |

Note: C represents partial sill, $C_0$ represents nugget, $C_0/(C_0 + C)$ represents space structure ratio, error represents root mean square standardized error.

Figure 5 shows the spatial distribution characteristics of land use diversity degree in Bayan County in 1991, 2005 and 2019, respectively. In 1991, the degree of land use diversity changed between 0 and 0.63, and the average value was 0.27. The proportion of land area with land use diversity degree below 0.31 was 78.35%, and the proportion of land area with land use diversity degree above 0.3 was 21.65%. In 2005, the degree of land use diversity ranged from 0 to 0.63, and the average value was 0.25. The percentage of land area with land use diversity degree below 0.30 was 78.52%. The percentage of land area with land use diversity degree above 0.30 was 21.48%. In 2019, the degree of land use diversity ranged from 0 to 0.61, and the average value was 0.26. The percentage of land area with land use diversity degree below 0.30 was 78.77%. The percentage of land area with land use diversity degree above 0.30 was 21.23%. The land use diversity degree was relatively high in the northwest, south and east of the study area, and relatively low in western–central and northeast regions. There are significant regional differences in the complexity of land use type composition in the 4148 sample areas in 1991, 2005 and 2019.

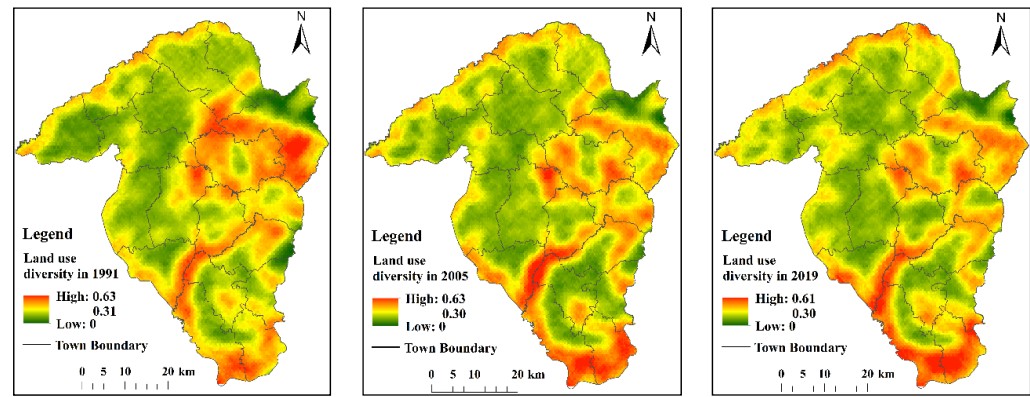

**Figure 5.** Spatial distribution characteristics of land use diversity degree.

### 3.1.2. Sample Line Identification Based on Land Use Diversity Degree

Figure 6 shows the layout scheme of the sample line based on the area with the average land use diversity degree in 1991, 2005 and 2019 (above 0.49) in the layer. The study area, whose average land use diversity degree was higher than 0.49, is covered with minimum sample transects (900 × 900 m). We selected 6630 consecutive samples in the study area as the sample belts of LUP changes, and the center line of sample belt was picked up as the sample line. The length of the sample line is about 189.9 km (6330 × 30 m). A new method for obtaining sample lines is proposed. Compared with previous sampling methods [51,52], the land use diversity degree of the base map ensured that the selected sample lines could include as many land use types as possible.

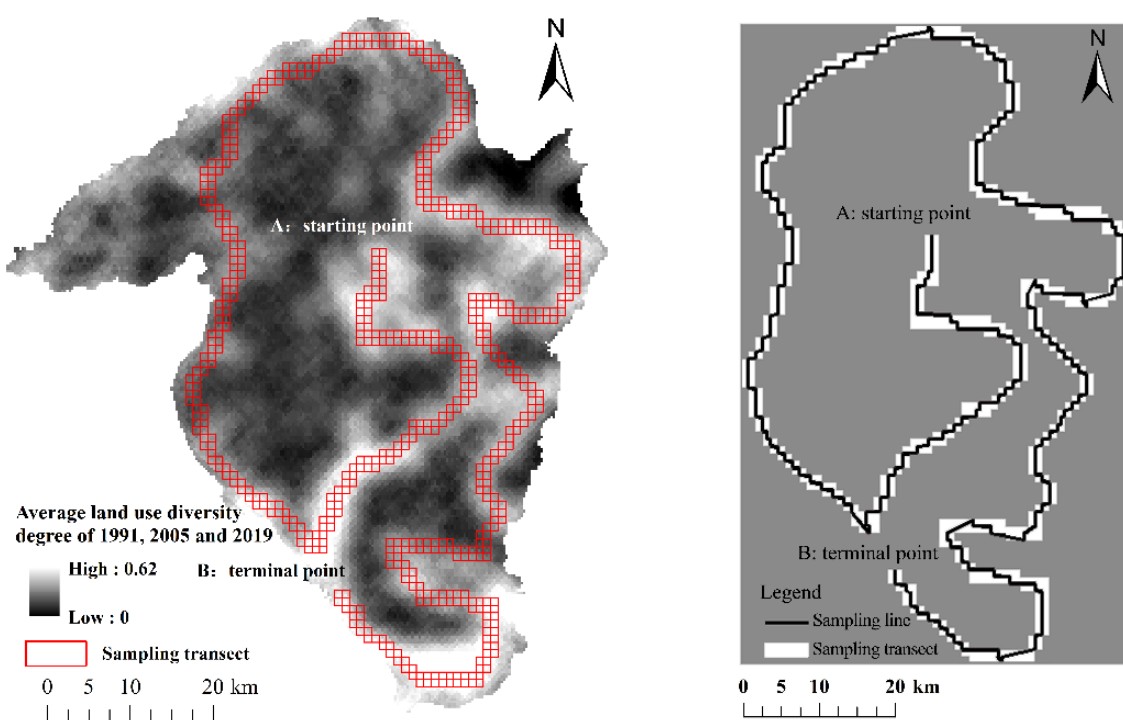

**Figure 6.** Sampling transect and sampling line.

### 3.2. Identify the Interaction Grid Scales between LUPs and SEFs

Figure 7 shows the interaction grid scales between LUPs and SEFs, and the data in Figure 7 were taken from the longitudinal coordinate's value corresponding to the data which passed the standard spectrum test of red noise in the XWT spectrum (Appendix A, Figure A1). At least two interaction grid scales could explain the impact of LUPs with population, GDP and land use intensity in 1991, 2005 and 2019, respectively. This article shows the length of a side of the interaction grid scales in Table 2.

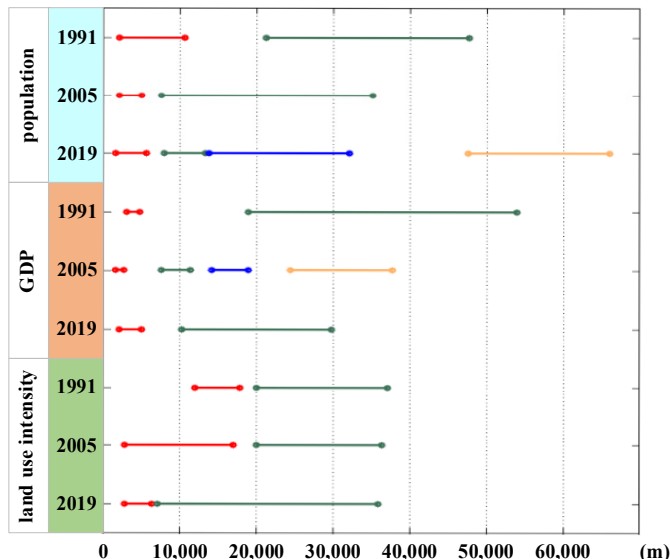

**Figure 7.** The interaction grid scales of LUPs with SEFs.

**Table 2.** The values of grid scales showing interaction between LUPs and SEFs (unit: m).

| Factors | Years | Interaction Grid Scales | | | |
|---|---|---|---|---|---|
| | | **(1)** | **(2)** | **(3)** | **(4)** |
| Population | 1991 | 1984–10,591 | 21,181–47,644 | / | / |
| | 2005 | 1983–5008 | 7486–35,120 | / | / |
| | 2019 | 1574–5595 | 7913–13,208 | 13,694–32,094 | 47,548–65,992 |
| GDP | 1991 | 2978–4758 | 18,820–53,975 | / | / |
| | 2005 | 1574–2629 | 7486–11,259 | 14,136–18,820 | 4400–37,641 |
| | 2019 | 1984–5008 | 10,226–29,738 | / | / |
| Land use intensity | 1991 | 11,911–17,805 | 20,032–37,122 | / | / |
| | 2005 | 2798–16,962 | 19,992–36,358 | / | / |
| | 2019 | 2805–6295 | 7082–35,858 | / | / |

The interaction grid scales of LUPs with population are 1984–10,591 m and 21,181–47,644 m in 1991, 1983–5008 m and 7486–35,120 m in 2005, and 1574–5595 m, 7913–13208 m, 13,694–32,094 m and 47,548–65,992 m in 2019, respectively. Therefore, the influence laws between population and LUP could be chosen by the interaction grid scales of 1984 m, 1983 m and 1574 m to reveal changes in 1991, 2005 and 2019, respectively. The interaction grid scales from 7500 m to 35,000 m, or from 47,700 m to 66,000 m, could also be selected.

What is more, the interaction grid scales of LUPs with GDP are 2978–4758 m and 18,820–53,975 m in 1991, 1574–2629 m, 7486–11,259 m, 14,136–18,820 m and 24,400–37,641 m in 2005, and 1984–5008 m and 10,226–29,738 m in 2019, respectively. Accordingly, the minimal interaction grid scales (1574 m, 1984 m and 2978 m) are chosen to explain the influence law of GDP on the LUPs. Furthermore, the interaction grid scales between 10000 m and 37,000 m could be used to reveal the influence laws between GDP and LUPs.

In addition, the interaction grid scales of LUPs with land use intensity are 11,911–17,805 m and 20,032–37,122 m in 1991, 2798–16,962 m and 19,992–36,358 m in 2005, and 2805–6295 m and 7082–35858 m in 2019, respectively. Thus, the interaction grid scales of 11911 m, 2798 m and 2805 m are selected to reveal the influence laws between land use intensity and LUPs in 1991, 2005 and 2019, respectively. Besides, the interaction grid scales between 19,000 m and 38,000 m could be used to explain the influence laws between land use intensity and LUPs.

The results indicate that the interaction grid scales of LUPs with population, GDP and land use intensity are basically distributed between 1500 and 66,000 m. In the ranges of 2978–5008 m and 24,400–29,738 m, the interaction grid scales between the LUPs and SEFs (population, GDP and land use intensity) from 1991 to 2019 are overlapping. Interestingly, there is a big grid size (66,000 m) in the study that could cover almost the whole area of Bayan County. The results suggest that big grids also reveal the law of action between LUPs and SEFs in Bayan County. The interaction grid scales of LUPs with SEFs are not the smaller, the better. In previous studies, the selection of data sources was carried out according to grids and statistical data [53], and the statistical data analysis was usually not as accurate as grids in revealing the relationship between LUPs and the influencing factors [54]. In the selection of data sources on the basis of grids, many researchers were committed to refining the resolution to reveal the interaction of LUPs and influencing factors, while some studies found that the resolution was not better when more refined [55]. The XWT spectrum identified the interaction grid scales between LUPs and SEFs, avoiding the lack of artificial grid scales selection in previous studies.

### 3.3. The Interaction Direction between LUPs and SEFs

The phase angle of the WTC spectrum (Figure A2) was used to reveal the interaction direction between SEFs and LUPs, and it could be used to analyze the leading (or lagging) change relationship between SEFs and LUPs, and to explain the driving and feedback effects. If the phase angle was from $\pi/2$ to $3\pi/2$ (negative), the phase change of LUPs lagged behind the change of SEFs. That is to say that SEFs would lead to a change in the

LUPs. This means that SEFs have a driving role in LUPs. If the phase angle is between $-\pi/2$ and $\pi/2$ (positive), the phase change of LUPs is ahead of SEFs. Namely, the change of LUPs would cause a change of SEFs in turn. This implies that SEFs provide feedback on the LUPs.

Table 3 shows the locations of interaction direction of LUPs with population, GDP and land use intensity in the study area in 1991, 2005 and 2019. The interaction direction varies between the population and the LUPs on the location of the sample line. The locations of positive action directions between population and LUPs are 55,110–81,930 m in 1991, 66,600–84,120 m in 2005, and 89,880–113,400 m and 93,690–108,200 m in 2019. This shows that the change of population lagged behind the LUPs, while population had feedback effects on LUPs in the study area. The negative action directions between population and LUPs are 38,190–161,000 m in 1991, 5310–35,700 m in 2005, and 123,540–139,100 m and 134,460–154,710 m in 2019. This shows that the population dominated the LUPs, while population had driving effects on the LUPs in the study area.

**Table 3.** The locations of interaction direction of LUPs with SEFs (unit: m).

| Years Factors | Population | | GDP | | Land Use Intensity | |
|---|---|---|---|---|---|---|
| | Positive (+) | Negative (−) | Positive (+) | Negative (−) | Positive (+) | Negative (−) |
| 1991 | 55,110–81,930 / / | 38,190–161,000 / / | 56,490–93,420 / / | 152,250–161,900 111,500–148,900 41,160–158,280 | 18,450–25,290 56,760–66,900 / | 39,240–164,500 161,280–182,100 / |
| 2005 | 66,600–84,120 / / / | 5310–35,700 / / / | 67,340–95,340 / / / | 6420–25,020 121,600–142,400 106,600–174,150 / | / / / / | 8610–45,670 122,160–137,800 122,600–182,200 110,130–163,740 |
| 2019 | 89,880–113,400 93,690–108,200 / / | 12,3540–13,9100 13,4460–15,4710 / / | 93,990–105,200 73,440–85,230 163,740–176,600 / | 11,070–27,480 22,830–43,080 109,560–126,810 110,500–127,400 135,000–159,400 | 54,570–63,330 / / / | 6420–41,700 26,670–44,190 117,500–134,800 117,000–180,990 109,020–159,100 |

Moreover, the interaction direction between GDP and LUPs are significantly different on the sample line, and the distribution is relatively fragmented. The positive action directions between GDP and LUPs are 56,490–93,420 m in 1991, 67,340–95,340 m in 2005, and 93,990–105,200 m, 73,440–85,230 m and 163,740–176,600 m in 2019. This shows that the change of GDP lagged the LUPs, while GDP had feedback effects on LUPs in the study area. The negative action directions between GDP and LUPs are 152,250–161,900 m, 111,500–148,900 m and 41,160–158,280 m in 1991, 6420–25,020 m, 121,600–142,400 m and 106,600–174,150 m in 2005, and 11,070–27,480 m, 22,830–43,080 m, 109,560–126,810 m, 110,500–127,400 m and 135,000–159,400 m in 2019. This shows that GDP dominated the LUPs, while GDP had driving effects on the LUPs in the study area.

In addition, the interaction direction between land use intensity and LUPs is significantly different on the sample line. The positive action directions between land use intensity and LUPs are 18,450–25,290 m and 56,760–66,900 m in 1991, and 54,570–63,330 m in 2019. This shows that the change of land use intensity lagged the LUPs, while land use intensity had feedback effects on LUPs in the study area. The negative action directions between land use intensity and LUPs are 39,240–164,500 m and 161,280–182,100 m in 1991, 8610–45,670 m, 122,160–137,800 m, 122,600–182,200 m and 110,130–163,740 m in 2005, and 6420–41,700 m, 26,670–44,190 m, 117,500–134,800 m, 117,000–180,990 m and 109,020–159,100 m in 2019. This shows that land use intensity dominated the LUPs, while land use intensity had driving effects on the LUPs in the study area.

The data about the spatial location between SEFs and LUPs in Figure 8 are taken from the phase angle of sample line position (cross coordinate value) corresponding to the WTC spectrum. In Figure 8, the orange lines and the blue lines represent the positive

and negative actions between SEFs and LUPs, respectively. The positive actions mean that SEFs are affected by the changes of LUPs, while the negative actions indicate that SEFs play an obvious role in driving changes of LUPs. On the whole, the interaction directions between SEFs and LUPs from 1991 to 2019 are almost negative on all the sample lines, while the interaction directions in the middle sample line of population and GDP from 1991 to 2019, the end sample line of GDP in 2019, and the start sample line of land use intensity in 1991 are positive. These negative sample lines represent the spatial locations, which are the construction land in Bayan County with higher population, GDP and land use intensity. It is noteworthy that there are three significant long sample lines found in 1991, corresponding to the interaction directions of LUPs with population, GDP and land use intensity, respectively. These positive sample lines represent the spatial locations that are mountainous areas in northeast Bayan County with a lower population, GDP and land use intensity. Some studies had reported the interaction direction between SEFs and LUPs according to multivariable linear regression and geographical detectors, and found that the interaction direction was the same as that of the driving effect [56,57]. The results were not beneficial to the analysis of interaction direction between SEFs and LUPs due to the limitations of all positive or all negative effects [58,59]. In addition, there was an article found that showed GDP and population were negatively related with the cultivated land and water area, respectively, and they played a positive role in construction land. This means that the actions of SEFs on LUPs were not a single driving effect or feedback effect [60]; the results were similar to our findings. In the study, the positive and negative actions in the interaction direction between SEFs and LUPs were obtained by wavelet analysis, suggesting that the results obtained using wavelet analysis in this study were finer than those obtained by linear regression and geographical detector models.

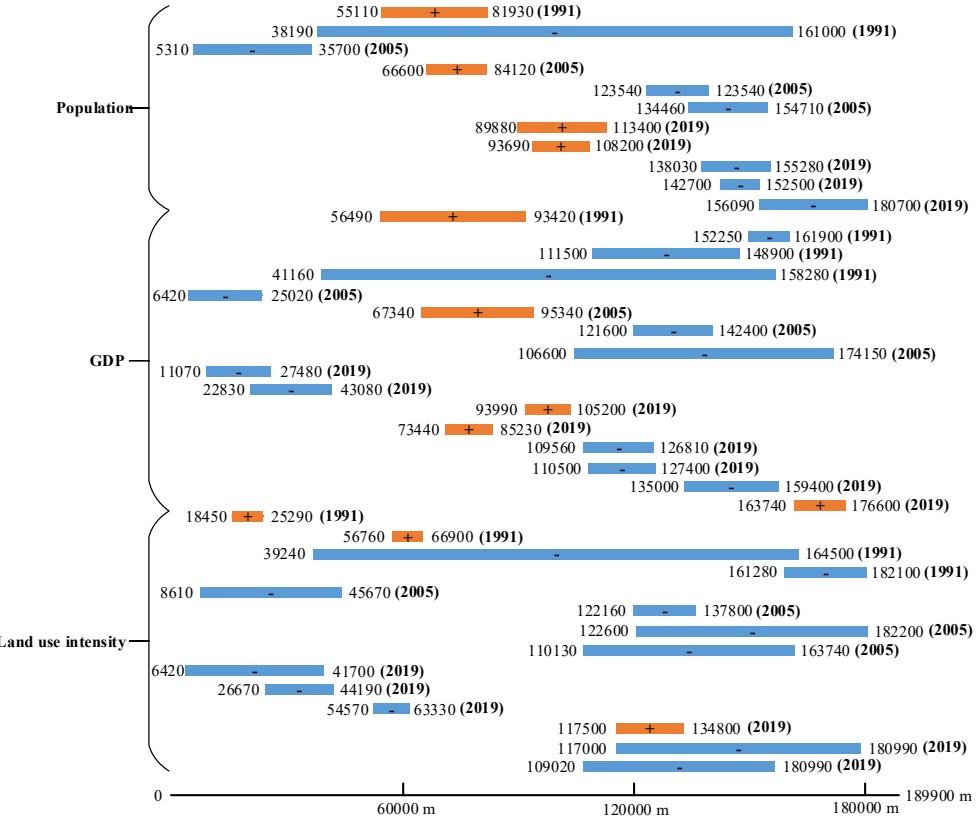

**Figure 8.** The interaction direction between SEFs and LUPs.

### 3.4. The Impact Strength Relationship between LUPs and SEFs

The WTC spectrum provides a useful method to better understand the causal relationships of the data, and it has advantages in analyzing the spatial influence of SEFs on

the LUPs. Table 4 shows the action strength of SEFs on the changes of LUPs based on the wavelet coherence coefficients (CS). Using WTC spectral density, this work analyzes the impact strength relationship between the LUPs and SEFs. The CS ranged from 0 to 1. The larger the CS, the stronger the effect of the SEFs on the LUPs. The lower the CS, the weaker the influence of the SEFs on the LUPs under the corresponding spatial scale.

**Table 4.** The CS and the number of samples tested by red noise (Numbers) of land use types, LUPs and SEFs.

| Land Use Types / Factors | | CS and Numbers | Dry Land | Paddy Field | Forest Land | Grass-land | Construction Land | Water Area | Other Land | LUPs |
|---|---|---|---|---|---|---|---|---|---|---|
| Population | 1991 | CS | **0.8236** | 0.8166 | 0.8046 | **0.8367** | 0.8026 | 0.8169 | **0.8358** | 0.8213 |
| | | Numbers | 37617 | 3944 | 9145 | 1545 | 3757 | 1225 | 3772 | 61,005 |
| | 2005 | CS | **0.8256** | 0.7948 | 0.7878 | 0.8010 | **0.8227** | 0.8049 | 0.8163 | 0.8198 |
| | | Numbers | 30360 | 2919 | 9860 | 1610 | 3265 | 1249 | 3178 | 52,461 |
| | 2019 | CS | **0.8148** | 0.8269 | 0.7922 | 0.8245 | **0.8196** | 0.8184 | **0.8324** | 0.8133 |
| | | Numbers | 30859 | 7495 | 7902 | 834 | 3697 | 1015 | 3005 | 54807 |
| GDP | 1991 | CS | 0.8101 | 0.8022 | 0.7955 | 0.7827 | **0.8248** | 0.8122 | 0.8123 | 0.8139 |
| | | Numbers | 49,527 | 5507 | 16,875 | 2276 | 4642 | 2082 | 6349 | 87,258 |
| | 2005 | CS | **0.8103** | **0.8142** | 0.7652 | 0.7916 | **0.8121** | 0.7913 | 0.7022 | 0.8060 |
| | | Numbers | 49536 | 5513 | 16,592 | 2316 | 4582 | 1163 | 6515 | 87,366 |
| | 2019 | CS | 0.7989 | 0.7936 | 0.8006 | **0.8072** | **0.8084** | 0.7877 | 0.8021 | 0.8024 |
| | | Numbers | 49,517 | 5533 | 16,802 | 2279 | 4613 | 2011 | 6473 | 87,391 |
| Land use intensity | 1991 | CS | 0.8094 | 0.8045 | 0.8025 | 0.7992 | **0.8172** | 0.8133 | 0.8261 | 0.8168 |
| | | Numbers | 53,353 | 6946 | 20,119 | 3720 | 5488 | 2511 | 7334 | 99,471 |
| | 2005 | CS | **0.8193** | 0.7913 | 0.8012 | 0.8055 | **0.8198** | 0.7895 | 0.8037 | 0.8181 |
| | | Numbers | 49,133 | 73141 | 19,811 | 2598 | 6312 | 2347 | 5581 | 97,986 |
| | 2019 | CS | **0.8376** | 0.8231 | **0.8360** | **0.8377** | **0.8362** | 0.8326 | **0.8365** | 0.8340 |
| | | Numbers | 48,764 | 12,953 | 18,941 | 1549 | 7325 | 2252 | 4871 | 96,655 |

Notes: The bold data indicate the CS values of the LUPs larger than that of the overall land use pattern, and the corresponding land use types are the focus of this study.

The CS of population, GDP and land use intensity with LUPs shows that there was obvious interaction between SEFs and LUPs in the study area in 1991, 2005 and 2019, and their CS is mostly distributed around 0.8. The impact strength relationship between population and LUPs is analyzed by CS. First, the impact strength relationship between population and LUPs was reduced at the important time nodes of 1991, 2005 and 2019, and their CS are 0.8213, 0.8198 and 0.8133, respectively. The influence values showing the relationship between population and land use types beyond the above CS values are dry land (0.8236), grassland (0.8367) and other land (0.8358) in 1991, dry land (0.8256) and construction land (0.8227) in 2005, and dry land (0.8148), construction land (0.8196) and other land (0.8324) in 2019, respectively. Further, the CS values showing the relationship between GDP and LUPs are 0.8139, 0.8060 and 0.8024 in 1991, 2005 and 2019, respectively. The influence values showing the relationship between GDP and land use types beyond the above CS values are construction land (0.8248) in 1991, dry land (0.8103), paddy field (0.8142) and construction land (0.8121) in 2005, and grassland (0.8072) and construction land (0.8084) in 2019, respectively. Finally, the CS values showing the relationship between land use intensity and LUPs are increased in order at the important time nodes of 1991, 2005 and 2019, and the CS values are 0.8168, 0.8181 and 0.8340, respectively. The degree of influence between land use intensity and land use types beyond the impact strength is construction land (0.8172) in 1991, dry land (0.8193) and construction land (0.8198) in 2005, and dry land (0.8376), forest land (0.8360), grassland (0.8377), construction land (0.8362) and other land (0.8365) in 2019, respectively. The order in effects of SEFs on LUPs is shown as follows: dry land, grassland, construction land, paddy field, forest land, other land, and water area during 1991 to 2019 (Table 4). The impact of population and GDP on the LUPs was reduced from 1991 to 2019, while the impact of land use intensity on the LUPs increased. This indicates that the SEFs in the study area were mainly farming and industrial development, and SEFs in agricultural and industrial areas would have a significant impact on the change of LUPs. However, SEFs were also gradually separated from the restriction of "land elements". Some studies had shown that the majority of the changes of LUPs

were caused by the changes of natural factors or major policy changes [61–63], while there were fewer major changes in natural conditions or policies and planning in the study area. SEFs were important factors that affected the LUPs in the study area [64]. Furthermore, the results also revealed that most SEFs in the study area related to the cultivated land and construction land, since the proportion of black soil area reached 53.5% of the total study area and 81.3% of the total cultivated land area with excellent farming conditions. The state regarded the study area as a commodity grain-base county and a demonstration area of high-standard farmland. In addition, being the suburban county of Harbin (the capital of Heilongjiang Province), the region was a provincial industrial demonstration base and had great advantages in the pork on the hoof, corn and soybean deep processing industries, leading to their frequent attraction of investment, real estate development, and the significant impact of SEFs in the region on construction land.

In the transfer of land use types, most of the construction land development and construction projects actually occupied more obvious grassland except for a small amount of construction land occupied by cultivated land, indicating that the impact of SEFs on grassland was also based on the needs of socioeconomic development, rather than ecological protection. The above reasons are consistent with the results that dry land, grassland and construction land were affected by SEFs (Table 4). Some studies have used the regression analysis model to determine the relationship between LUPs and their influencing factors, and the relationship calculated by this linear model could not identify the internal minor changes [27,65]. In short, population, GDP and land use intensity were the important SEFs in the changes of LUPs, and were important driving forces for social progress and economic development in the study area. Previous studies have confirmed that different influencing factors will show different types of action in land use space [23,60]. Wavelet analysis can more accurately distinguish the local actions between LUPs and SEFs in different land use spaces.

## 4. Conclusions

Exploring the interaction relationship of LUP changes with population, GDP and land use intensity could help to reveal the evolution laws of land use change and provide a basis for future regional development planning and decision making. This work proposes a new sample line selection method using models of XWT and WTC. The appropriate multi-scale grids size, interaction direction and action strength of the interaction relationship between LUPs and SEFs were calculated by XWT and WTC models. There were two ranges, of 2978–5008 m and 24,400–29,738 m, in which the interaction grid scales between LUPs and SEFs (population, GDP and land use intensity) from 1991 to 2019 were overlapping. The interaction direction between LUPs and SEFs from 1991 to 2019 was almost negative on all sample lines, while the interaction directions in the middle sample line of population and GDP from 1991 to 2019, the end sample line of GDP in 2019, and the start sample line of land use intensity in 1991 were positive. Dry land, grassland and construction land were most affected by SEFs, follow by paddy fields, forest land and other land, and the least affected was water area from 1991 to 2019. The impact of population and GDP on LUPs was reduced from 1991 to 2019, while the impact of land use intensity on LUPs was increased.

The advantages of this study are that the spatial grid range suitable for the analysis of the interaction relationship between LUPs and SEFs was obtained at the county scale, which provided a new technical means to make up for the current deficiency of spatial grid size determination. In addition, this study used the wavelet analysis model to determine the driving and feedback relationship between LUPs and SEFs, believing that even the impact of the same influence factor on the land use pattern in the same period has complex positive and negative discrepancies. The method established in the study provides a new tool for the study of action relationships, and solves the deficiency in previous studies that found the interaction direction between LUPs and SEFs was only a one-way effect. This study could help to expand the understanding of interaction grid size dependence and provide a reference to decision makers for them to improve regional socioeconomic

development by regulating human activities. Deep exploration of the influence of policies and regulations, planning, tourism, culture and other social and economic factors on land use patterns will be the research focus and direction of the next step.

**Author Contributions:** Conceptualization, Y.W. and G.S.; methodology, Y.W. and W.L.; software, Y.W.; validation, Y.W.; formal analysis, Y.W.; resources, Y.W.; data curation, Y.W.; writing—original draft preparation, Y.W.; writing—review and editing, G.S.; visualization, Y.W. and W.L.; supervision, Y.W.; project administration, G.S.; funding acquisition, Y.W. and G.S. All authors have read and agreed to the published version of the manuscript.

**Funding:** This research was funded by the National Natural Science Foundation of China (grant numbers 41571165, 41971247), and the Scientific Research Fund of Liaoning Provincial Education Department, China (grant number WQN202016).

**Institutional Review Board Statement:** Not applicable.

**Informed Consent Statement:** Not applicable.

**Data Availability Statement:** The data presented in this study are available on request from the corresponding author.

**Acknowledgments:** We are thankful for the data support from the United States Geological Survey (USGS) (http://Landsat.usgs.gov/, accessed on 21 September 2021), Subject Database of Special Human-Land System for Informatization Construction of Chinese Academy of Sciences, and the *Statistical bulletin of national economic and social development of Bayan County, Heilongjiang Province (2019).*

**Conflicts of Interest:** The authors declare no conflict of interest.

**Appendix A**

**Table A1.** Abbreviations list.

| Abbreviations | Full Name (in Full) |
| --- | --- |
| LUPs | Land use patterns |
| SEFs | Socioeconomic factors |
| GDP | Gross domestic product |
| CWT | Continuous wavelet transform |
| XWT | Cross wavelet transform |
| WTC | Wavelet transform coherence |

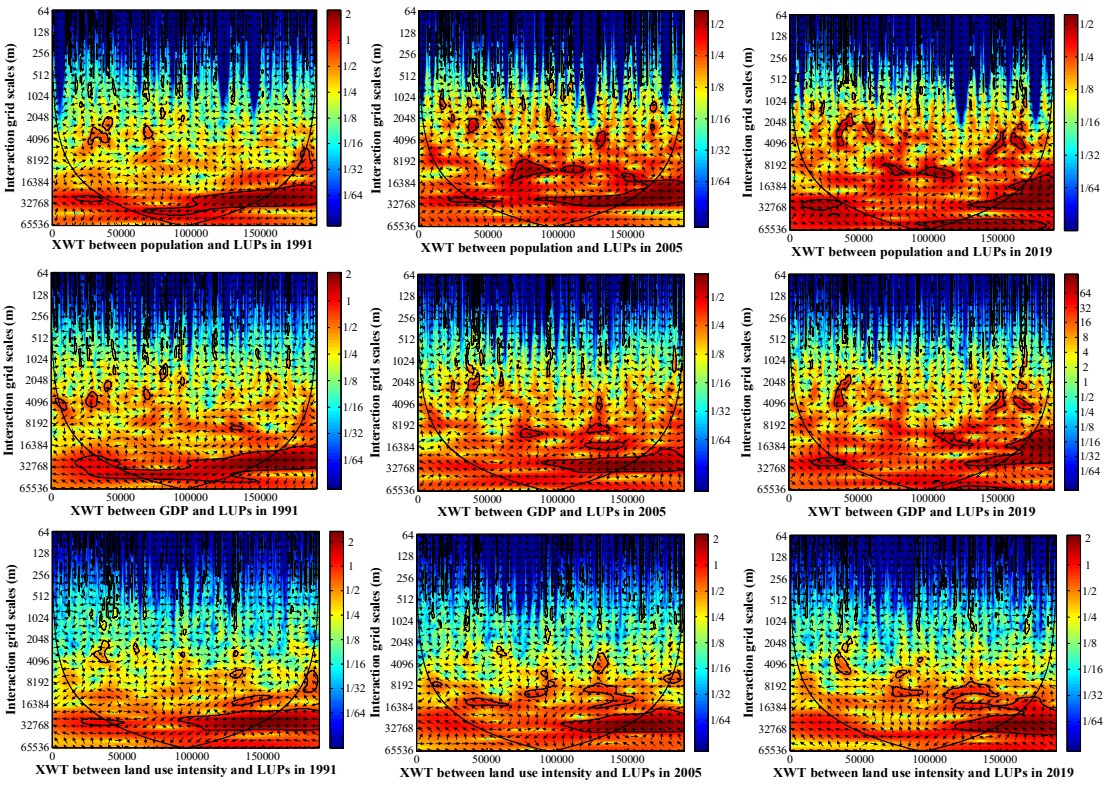

**Figure A1.** The XWT between SEFs and LUPs in 1991, 2005 and 2019.

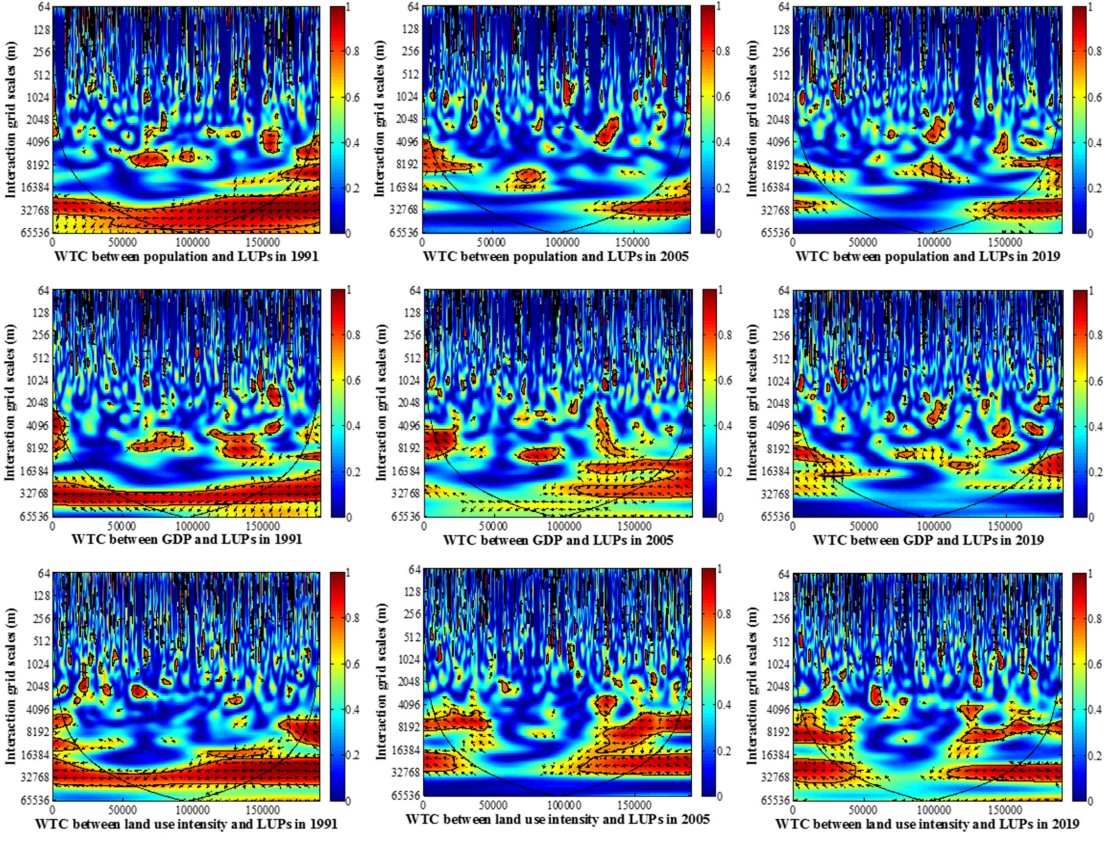

**Figure A2.** The WTC between SEFs and LUPs in 1991, 2005 and 2019.

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
