# Peer review of "The Interaction Relationship between Land Use Patterns and Socioeconomic Factors Based on Wavelet Analysis: A Case Study of the Black Soil Region of Northeast China"

_land, doi:10.3390/land10111237_

Round 1
Reviewer 1 Report
the paper focuses on the interplay between socio-economic factors and land use in Bayan County in China. The paper is built with a quantitative approach that seems solid and supported by an appropriate use of cartography and graphics. The bibliography is coherent and updated and the scientific soundness is adequate.
Author Response
Thank you for your valuable comments, which are of great help and significance to the improvement of this research. We have carefully considered and revised the manuscript according to your suggestions. And we believe that the revisions and adjustments have substantially improved our manuscript (ID: land- 1410045).
Point: The paper focuses on the interplay between socio-economic factors and land use in Bayan County in China. The paper is built with a quantitative approach that seems solid and supported by an appropriate use of cartography and graphics. The bibliography is coherent and updated and the scientific soundness is adequate.
Response: Thanks to the reviewer’s affirmation of this article. We make some modifications and mark in red in the revised manuscript.
Please see the attachment.

Reviewer 2 Report
The manuscript is well structured, data used are appropriate, however I am not familliar with cross wavelet transform, so I can not judge about its use (i am using for such data canonical ordination).
It is not clear what is the main idea of the paper - if it is methodological paper, the mothod must be better describet in Methods section and discusses deeply i n Discussion part. If it is empirical manuscript - data are of authors main objective the discussion must be enlarged and deeply dicsussed what was found in the arms of land use changes and regional development planning in China.
Author Response
Dear reviewer,
Thank you for your valuable comments, which are of great help and significance to the improvement of this research. We have carefully considered and revised the manuscript according to your suggestions. And we believe that the revisions and adjustments have substantially improved our manuscript (ID: land- 1410045).
Point: The manuscript is well structured, data used are appropriate, however I am not familliar with cross wavelet transform, so I can not judge about its use (i am using for such data canonical ordination).
It is not clear what is the main idea of the paper - if it is methodological paper, the mothod must be better describet in Methods section and discusses deeply in Discussion part. If it is empirical manuscript - data are of authors main objective the discussion must be enlarged and deeply dicsussed what was found in the arms of land use changes and regional development planning in China.
Response: Thank you very much for your suggestion. According to your suggestions, we revisit the wavelet analysis methods in ‘2.3 Model’ (in lines 191-201) and ‘2.3.1 Continuous wavelet transform (CWT)’ (in lines 202-219). We strengthen the discussion of the method innovation and the related contents of land use changes and regional development planning as follows:
(1) The XWT spectrum identified the interaction grid scales between LUPs and SEFs, avoiding the lack of artificial grid scales selection in previous studies (in lines 385-387).
(2) In addition, there was an article found that GDP and population showed negatively related with the cultivated land and water area, respectively, and they played a positive role in construction land. It meant that the actions of SEFs on LUPs were not a single driving effect or feedback effect [61], the results were similar to our findings (in lines 452-456).
(3) Previous studies have confirmed that different influencing factors will show different ways of action in land use space [23, 61]. Wavelet analysis can more accurately distinguish the local actions between LUPs and SEFs in different land use spaces (in lines 524-527).
Please see the attachment.

Reviewer 3 Report
The paper deals with an interesting and important topic.
Nevertheless, I recommend an extensive revision before it can be published.
Here some comments for the revision:
- The document contains a lot of abbreviations. Therefore I recommend to add a list of abbreviations. Please check if all abbreviations are really necessary (e.g. POP for Population). The less abbreviations, the easier it is to read the text.
- The CWT method needs to be better explained. It must also be explained why it is used here.
- Figure 2 is not readable (too small). Please explain what exactly the maps say. What differences can be derived? Why were these annual dates chosen for the analysis? Overall (for all maps the significance has to be improved/increased)
- Chapter 2.3: This chapter is difficult to read. Further explanation is needed here for better understanding.
- Chapters 3.2 and 3.3: These chapters are difficult to read. Wouldn't a table containing all the data be better here?
- Figure 6 is not explained in the text. A reference is missing. Without explanation it is not understandable. Otherwise, the figure can also be removed.
- The same applies to Figure 8 (in addition, the figure caption is in the wrong place). What is the content of the figures?
Author Response
Dear reviewer,
Thank you for your valuable comments, which are of great help and significance to the improvement of this research. We have carefully considered and revised the manuscript according to your suggestions. And we believe that the revisions and adjustments have substantially improved our manuscript (ID: land- 1410045).
All the changes are marked red fonts in the revised version. The specific revision and instructions are as follows:
Point 1: The document contains a lot of abbreviations. Therefore I recommend to add a list of abbreviations. Please check if all abbreviations are really necessary (e.g. POP for Population). The less abbreviations, the easier it is to read the text.
Response 1: Thank you very much for your suggestion. According to your suggestions, we remove the abbreviation for POP and LUI. There are six abbreviations appearing in this article, LUPs, SEFs, GDP, CWT, XWT and WTC represents land use pattern, socioeconomic factors, gross domestic product, continuous wavelet transform, cross wavelet transform and wavelet transform coherent, respectively. The list of abbreviations are shown in the appendix 1.
Point 2: The CWT method needs to be better explained. It must also be explained why it is used here.
Response 2: Thank you very much for your suggestion. The basic theory of continuous wavelet transform (CWT) has been presented in this paper before using XWT and WTC. CWT could achieve multi-scale refinement of localized information of temporal (spatial) frequency, and automatically could adapt to the multi-scale identification requirements of interactions between LUPs and SEFs (in lines 203-205).
Point 3: Figure 2 is not readable (too small). Please explain what exactly the maps say. What differences can be derived? Why were these annual dates chosen for the analysis? Overall (for all maps the significance has to be improved/increased)
Response 3: Thank you very much for your suggestion. Figure 2 shows that the area of land use types is characterized by the gradual increase of cultivated land (dry land and paddy field) and construction land, and by the decrease of forest land, grassland and other land from 1991 to 2019. The water area was basically relatively stable from 1991 to 2019 (in lines 150-153).
This research selects 1991 and 2019 as the study period of LUPs, and analyzes the relationship between the LUPs change and SEFs during promulgation of the Land Administration Law Implementation Regulations in 1991 and the second revision of the China Land Administration Law in 2019. The year 2005 is used as the intermediate node for 1991 and 2019 as a reference to avoid bias caused by stochastic trends in the starting time nodes (in lines 129-134).
We improve the accuracy of all the maps, including Figure 1, Figure 2, Figure 3, Figure 5 and Figure 6.
Point 4: Chapter 2.3: This chapter is difficult to read. Further explanation is needed here for better understanding.
Response 4: Thank you very much for your suggestion. According to your suggestions, we explained the application of the wavelet analysis in this article as follows: The basic theory of continuous wavelet transform (CWT) will be presented in the study before using cross wavelet transform (XWT) and wavelet transform coherent (WTC). XWT could be used to explore the interaction grid scales between LUPs and SEFs [47]. WTC was used to measure the degree of local correlation between LUPs and SEFs. The phase angle of WTC spectrum was used to reveal the interaction direction between SEFs and LUPs. We also added a study framework in Figure 4 (in lines 191-196 and 291).
Point 5: Chapters 3.2 and 3.3: These chapters are difficult to read. Wouldn't a table containing all the data be better here?
Response 5: Thank you very much for your suggestion. According to your suggestions, we have supplemented Table 2 and Table 3 in line 371 and line 431.
Point 6: Figure 6 is not explained in the text. A reference is missing. Without explanation it is not understandable. Otherwise, the figure can also be removed.
Response 6: Thank you very much for your suggestion. According to your suggestions, we have removed the original Figure 6, and put them as an attachment in the appendix 2.
Point 7: The same applies to Figure 8 (in addition, the figure caption is in the wrong place). What is the content of the figures?
Response 7: Thank you very much for your suggestion. According to your suggestions, we have already removed the original Figure 8. This Figure is uploaded as an attachment in the appendix 3.
Please see the attachment.

Reviewer 4 Report
This paper is presented as an interesting and current study, as the subject of urban land use is approached more and more often, especially in the context of economic change, population growth in urban areas and implicitly of urban areas, trying to find solutions for a sustainable regional land utilization and sustainable social and economic development
I appreciate the approach of the subject, which combines remote sensing with statistical methods and methods of spatial representation, which facilitates the understanding of the discussed phenomenon.
However, I believe that for a better understanding of this work some modifications are necessary, which I will highlight below. My analysis was done by chapters, but my remarks will only focus on part of them.
Introduction: This chapter respects the scientific character of such a paper, the general topic of the article and of the study area being outlined and explained by a sufficient number of quotations. (Citations).
Material and methods: This chapter is well structured and explained. However, I recommend inserting a workflow scheme for easier tracking of the methodology.
It would be advisable to insert in the maps in the introduction some toponyms (city names). Those who do not know the study area (they are not from that area) can look for additional information based on this (of the cities in your study area).
Results and discussions : This chapter highlights an editing issue. Most likely the name of figure 8 is placed above the illustration creating a state of confusion for the reader.
Try to use the present tense in the expression: "Fig. 8 showed the interaction direction ..." => "Fig. 8 shows the interaction direction .."
Figure 8 is placed in the middle of a paragraph.
Conclusions: This last chapter treats very well, in a compact and easy to understand way the complex study that has been done. From my point of view, nothing should be changed in this regard.
In conclusion, the study is a well-developed one and from my own perspective minor changes should be made only to the Material and methods and Results and discussions chapters, as I wrote above.
Author Response
Dear reviewer,
Thank you for your valuable comments, which are of great help and significance to the improvement of this research. We have carefully considered and revised the manuscript according to your suggestions. And we believe that the revisions and adjustments have substantially improved our manuscript (ID: land- 1410045).
All the changes are marked red fonts in the revised version. The specific revision and instructions are as follows:
Point 1: Introduction: This chapter respects the scientific character of such a paper, the general topic of the article and of the study area being outlined and explained by a sufficient number of quotations. (Citations).
Response 1: Thank you for your suggestion. According to your suggestions, we add six literature articles (References: 20-25) to supplement the research trend. Many researches have been conducted on the effect of SEFs on LUPs [20-26]. Dong et al. (2021) investigated the spatiotemporal patterns and driving factors of land use and land cover change in the China-Mongolia-Russia Economic Corridor, and found that SEFs were more important to the change of LUPs than to natural factors [22]. Zhou et al. (2020) considered the driving factors of land use change in rural China from 1995 to 2015, and discovered that socioeconomic development was the main driving force of construction land expansion [23]. Qie et al. (2017) established a spatial-temporal human exposure model based on land use changes in Dalian City (China) at a city scale, and found that SEFs were the contributors to the spatial distribution variance of land use changes [26]. Liu et al. (2020) reported the impacts of SEFs on land use changes in Mekong River from a watershed scale over the last 40 years, and found that SEFs such as economic development and land policies were the main driving forces of land use changes [27]. Previous studies had shown that SEFs were important driving factors of LUPs. Previous results on the effects of SEFs on the changes of LUPs were not always coincident due to the difference of spatial range in various study areas [28-31]. However, SEFs generally had both positive and negative effects on LUPs, which was less studied in previous studies (in lines 64-81).
Point 2: Material and methods: This chapter is well structured and explained. However, I recommend inserting a workflow scheme for easier tracking of the methodology.
Response 2: Thank you for your suggestion. According to your suggestions, we add a study framework in line 291.
Point 3: It would be advisable to insert in the maps in the introduction some toponyms (city names). Those who do not know the study area (they are not from that area) can look for additional information based on this (of the cities in your study area).
Response 3: Thank you for your suggestion. According to your suggestions, we have supplemented the names of the 18 towns in Figure 1 (in lines 146-147).
Point 4: Results and discussions: This chapter highlights an editing issue. Most likely the name of figure 8 is placed above the illustration creating a state of confusion for the reader.
Response 4: Thank you very much for your suggestion. According to your suggestions, we have already removed the original Figure 8. This Figure is uploaded as an attachment in the appendix 3.
Point 5: Try to use the present tense in the expression: "Fig. 8 showed the interaction direction ..." => "Fig. 8 shows the interaction direction."
Response 5: Thank you very much for your suggestion. According to your suggestions, we have changed the expression by present tense in Figure 8, and similarly, the result expression of other figures and tables have been modified to present tense.
Point 6: Figure 8 is placed in the middle of a paragraph.
Response 6: Thank you very much for your suggestion. We have already removed the original Figure 8. This Figure is uploaded as an attachment in the appendix 3.
Point 7: Conclusions: This last chapter treats very well, in a compact and easy to understand way the complex study that has been done. From my point of view, nothing should be changed in this regard.
Response 7: Thank you very much for your affirmation. We only modify the tense, grammar, and acronyms of this section.
Please see the attachment.

Round 2
Reviewer 2 Report
All my comments were considered by authors.